# Simple Hardware-Efficient PCFGs with Independent Left and Right Productions

**Wei Liu**[1,2*]**, Songlin Yang**[3*]**, Yoon Kim**[3]**, Kewei Tu**[1,2]

[1]School of Information Science and Technology, ShanghaiTech University
[2]Shanghai Engineering Research Center of Intelligent Vision and Imaging
[3]Massachusetts Institute of Technology
{liuwei4,tukw}@shanghaitech.edu.cn
{yangsl66,yoonkim}@mit.edu

## Abstract

Scaling dense PCFGs to thousands of nonterminals via a low-rank parameterization of the rule probability tensor has been shown to be beneficial for unsupervised parsing. However, PCFGs scaled this way still perform poorly as a language model, and even underperform similarly-sized HMMs. This work introduces *SimplePCFG*, a simple PCFG formalism with independent left and right productions. Despite imposing a stronger independence assumption than the low-rank approach, we find that this formalism scales more effectively both as a language model and as an unsupervised parser. As an unsupervised parser, our simple PCFG obtains an average F1 of 65.1 on the English PTB, and as a language model, it obtains a perplexity of 119.0, outperforming similarly-sized low-rank PCFGs. We further introduce *FlashInside*, a hardware IO-aware implementation of the inside algorithm for efficiently scaling simple PCFGs.

## 1 Introduction

Despite the improvements in unsupervised parsing obtained through scaling neural probabilistic context-free grammars (PCFGs), their language model performance scales less favorably compared to, for example, hidden Markov models (HMMs) and neural language models. On the Penn Treebank, a neural PCFG with 30 nonterminals and 60 preterminals obtains $\approx 250$ perplexity (Kim et al., 2019), and while scaling neural PCFGs to thousands of states via a low-rank parameterization can improve perplexity to $\approx 170$ (Yang et al., 2022), this still lags behind a similarly-sized HMM, which obtains $\approx 130$ perplexity (Chiu et al., 2021), despite the fact that HMMs are a subclass of PCFGs

This work proposes *SimplePCFG*, a simple PCFG formalism with independent left and right productions. We find that this simple PCFG scales

more effectively (in terms of both language modeling and unsupervised parsing) than previous approaches which scale PCFGs by factorizing the rule probability tensor into low-rank components (Yang et al., 2021b, 2022). In particular, we find that simple PCFGs can obtain significantly lower perplexity in language modeling while achieving higher unsupervised parsing performance compared to low-rank PCFGs with a similar number of nonterminals, achieving a near state-of-the-art unsupervised parsing performance on the Penn Treebank with an F1 of 65.1. We further describe a hardware-efficient IO-aware implementation of the inside algorithm, dubbed *FlashInside*, to facilitate scalable learning of simple PCFGs.

## 2 Simple PCFGs

A PCFG can be defined by a 6-tuple $\mathcal{G} = (S, \mathcal{N}, \mathcal{P}, \Sigma, \mathcal{R}, \pi)$, where $S$ is the distinguished start symbol, $\mathcal{N}/\mathcal{P}/\Sigma$ are a finite set of nonterminal/pre-terminal/terminal symbols,[1] $\mathcal{R}$ is a set of production rules of the form,

$$
\begin{aligned}
S &\to A, & A &\in \mathcal{N} \\
A &\to BC, & A &\in \mathcal{N}, B, C \in \mathcal{N} \cup \mathcal{P} \\
T &\to w, & T &\in \mathcal{P}, w \in \Sigma
\end{aligned}
$$

and $\pi : \mathcal{R} \to [0, 1]$ maps rules to their associated probabilities. In simple PCFGs, we decompose $\pi_{A \to BC}$ into $\pi_{B \curvearrowleft A} \cdot \pi_{A \curvearrowright C}$, effectively assuming that left and right children are generated independently.[2] We denote $\mathbf{L}, \mathbf{R} \in \mathbb{R}^{|\mathcal{N}| \times |\mathcal{N}|}$ as the matrix representation of $\pi_{B \curvearrowleft A}$ and $\pi_{A \curvearrowright C}$, and apply a neural parameterization over these matrices to compute the rule probabilities (Kim et al., 2019). See Appendix A for details.

---

*Equal contribution.
Code: https://github.com/sustcsonglin/TN-PCFG.

[1]For brevity we do not distinguish between $\mathcal{N}$ and $\mathcal{P}$ for the rest of the paper.

[2]This formalism has been discussed in Hsu et al. (2012), i.e., PCFG-I. We also experimented with PCFG-IE, where $\mathbf{L} = \mathbf{R}$, but found it necessary to distinguish between $\mathbf{L}$ and $\mathbf{R}$ to achieve good performance.

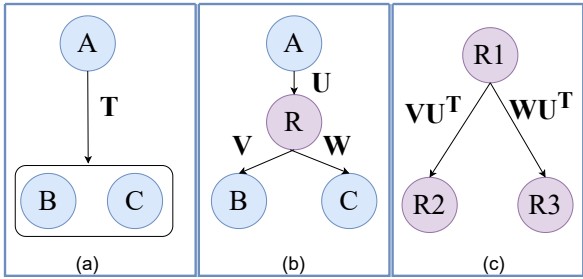

Figure 1: Bayesian network-like representations of PCFG binary rules: (a) original grammar, (b) after tensor decomposition (Yang et al., 2021b), and (c) rank space grammar (Yang et al., 2022). Our simple PCFG is almost the same as (c) but uses a flexible parameterization.

**Comparing simple vs. low-rank PCFGs.** The previous approach to scaling HMMs and PCFGs to thousands of nontermals is parameterizing the rule probability tensor $\mathbf{T} \in \mathbb{R}^{|\mathcal{N}| \times |\mathcal{N}| \times |\mathcal{N}|}$ to be low-rank (Chiu et al., 2021; Yang et al., 2021b, 2022). Low-rank PCFGs can be viewed as introducing a new latent variable, namely a "rank variable" $R$, to decompose $\pi_{A \to BC}$ into $\sum_R \pi_{A \to R}\pi_{B \curvearrowright R}\pi_{R \curvearrowright C}$, as shown in Fig. 1, where the tensor/matrix representations of $\pi_{A \to BC}, \pi_{A \to R}, \pi_{B \curvearrowright R}, \pi_{R \curvearrowright C}$ are $\mathbf{T}, \mathbf{U}, \mathbf{V}, \mathbf{W}$, respectively. Yang et al. (2022, Sect. 4.2) show that a low-rank PCFG can be reparameterized as a simple PCFG with independent left/right productions by marginalizing nonterminal variables and viewing the rank variables as new nonterminal variables. As such, low-rank PCFGs parameterize $\mathbf{L}, \mathbf{R}$ in a more restrictive manner: $\mathbf{L} = \mathbf{V}\mathbf{U}^T, \mathbf{R} = \mathbf{W}\mathbf{U}^T$. We speculate that the shared $\mathbf{U}^T$ would restrict the expressiveness of low-rank PCFGs and thus hinder optimization, which motivates our simple PCFGs.

## 3 A Hardware-efficient Inside Algorithm

### 3.1 The inside algorithm for simple PCFGs

The inside algorithm for simple PCFGs has the following recursive formula:

$$\beta_{ij}^A = \sum_{B,C \in \mathcal{N}} \pi_{B \curvearrowright A} \cdot \pi_{A \curvearrowright C} \sum_{i<k<j} \beta_{ik}^B \cdot \beta_{kj}^C$$

$$= \sum_{i<k<j} \underbrace{\left( \sum_{B \in \mathcal{N}} \pi_{B \curvearrowright A} \cdot \beta_{ik}^B \right)}_{\eta_{ik}^A} \underbrace{\left( \sum_{C \in \mathcal{N}} \pi_{A \curvearrowright C} \cdot \beta_{kj}^C \right)}_{\zeta_{kj}^A}$$

where $\beta_{ij}^A$ is the inside probability for span $(A, i, j)$ with the base case $\beta_{ii}^A = \pi_{A \to w_i}$. We cache $\eta_{ij}^A, \zeta_{ij}^A$

to avoid repeated computation, similar to Cohen et al. (2013) and Yang et al. (2022). The resulting complexity is $\mathcal{O}(l^3|\mathcal{N}| + l^2|\mathcal{N}|^2)$ where $l$ is sentence length.

**Vector form.** We abuse the notation to have $\boldsymbol{\beta}_{ij}, \boldsymbol{\eta}_{ij}, \boldsymbol{\zeta}_{ij} \in \mathbb{R}^{|\mathcal{N}|}$. Then we can write $\boldsymbol{\beta}_{ij} = \sum_{i<k<j} \boldsymbol{\eta}_{ik} \odot \boldsymbol{\zeta}_{kj}$ and $\boldsymbol{\eta}_{ij} = \mathbf{L}\boldsymbol{\beta}_{ij}, \boldsymbol{\zeta}_{ij} = \mathbf{R}\boldsymbol{\beta}_{ij}$, where $\odot$ is the element-wise product.

### 3.2 FlashInside

It is necessary to implement the inside algorithm on GPUs efficiently to facilitate scaling of simple PCFGs. We introduce *FlashInside*, a hardware-efficient IO-aware implementation of the inside algorithm in the spirit of *FlashAttention* (Dao et al., 2022). *FlashInside* comprises of four main techniques:

**Span-level parallelism.** Given the span width $w$, the inside probability vector $\boldsymbol{\beta}_{i(i+w)}$ could be computed in parallel for different starting position $i$ (Yi et al., 2011, Sect. 4.2).

**The log-einsum-exp trick.** To improve numerical stability, it is common to use the "log-sum-exp" trick. For example, letting $\boldsymbol{o}_{ij} = \log \boldsymbol{\beta}_{ij}, \boldsymbol{a}_{ij} = \log \boldsymbol{\eta}_{ij}, \boldsymbol{b}_{ij} = \log \boldsymbol{\zeta}_{ij}$, we have

$$\boldsymbol{o}_{ij} = \boldsymbol{x}^\star + \log \sum_{i<k<j} \exp(\boldsymbol{a}_{ik} + \boldsymbol{b}_{kj} - \boldsymbol{x}^\star) \quad (1)$$

where $\boldsymbol{x}^\star = \max_{i<k<j}(\boldsymbol{a}_{ik} + \boldsymbol{b}_{kj}) \in \mathbb{R}^{|\mathcal{N}|}$. Using log-sum-exp could be expensive when computing $\boldsymbol{a}_{ij}$ and $\boldsymbol{b}_{ij}$, so we resort to the "log-einsum-exp" trick (Peharz et al., 2020, Sect. 3.2),

$$\boldsymbol{a}_{ij} = \boldsymbol{x}^\dagger + \log \left( \mathbf{L} \exp(\boldsymbol{o}_{ij} - \boldsymbol{x}^\dagger) \right)$$
$$\boldsymbol{b}_{ij} = \boldsymbol{x}^\dagger + \log \left( \mathbf{R} \exp(\boldsymbol{o}_{ij} - \boldsymbol{x}^\dagger) \right)$$

where $\boldsymbol{x}^\dagger = \max \boldsymbol{o}_{ij} \in \mathbb{R}$.[3] This allows us to leverage matrix multiplication operators, which are highly optimized on GPUs, to compute $\mathbf{L} \exp(\boldsymbol{o}_{ij} - \boldsymbol{x}^\dagger)$ and $\mathbf{R} \exp(\boldsymbol{o}_{ij} - \boldsymbol{x}^\dagger)$.

**Kernel fusion.** The above computation involves many element-wise operations and is thus *memory-bounded*. Loading and storing these vectors multiple times would cause significant IO-cost (Dao et al., 2022). We reduce the IO-cost by fusing these operations whenever possible. Concretely, when

---

[3]We abuse the notation for broadcasting vector-scalar addition/subtraction.

| Algorithm | $|\mathcal{N}|$ | $l$ | Speed | Memory |
|---|---|---|---|---|
| log-sum-exp | 512 | 20 | 1x | 100x |
| log-einsum-exp | 512 | 20 | 4.8x | 3x |
| FlashInside | 512 | 20 | 9.5x | 1x |
| log-einsum-exp | 8192 | 20 | 1x | 2x |
| FlashInside | 8192 | 20 | 6x | 1x |
| log-sum-exp | 512 | 40 | 1x | 50x |
| log-einsum-exp | 512 | 40 | 16x | 3x |
| FlashInside | 512 | 40 | 44x | 1x |
| log-einsum-exp | 8192 | 40 | 1x | 2.4x |
| FlashInside | 8192 | 40 | 39x | 1x |

Table 1: Ablation of how different elements of FlashInside contribute to speed and memory effiency, with different numbers of nonterminals ($|\mathcal{N}|$) and sentence lengths ($l$). For speed the regular log-sum-exp implementation from Torch-Struct (Rush, 2020) is the baseline, whereas for memory our FlashInside serves as the baseline.

computing $\exp(\boldsymbol{o}_{ij} - \boldsymbol{x}^{\dagger})$, we perform $\max, -, \exp$ in the same kernel that computes $\boldsymbol{o}_{ij}$; and we compute $\boldsymbol{a}_{ij}, \boldsymbol{b}_{ij}$ at once by

$$[\boldsymbol{a}_{ij}|\boldsymbol{b}_{ij}] = \boldsymbol{x}^{\dagger} + \log\left([\mathbf{L}|\mathbf{R}]\exp(\boldsymbol{o}_{ij} - \boldsymbol{x}^{\dagger})\right)$$

followed by fused element-wise $\log$ and addition operations.

**Recomputation.** While it possible to rely on automatic differentiation (AD) to backpropagate through the inside algorithm (Eisner, 2016), this can be memory-inefficient since AD would save *all* the intermediate results in the DP computation, which are not needed. For example, in Eq. 1 the partial differentiation between $\boldsymbol{o}_{ij}$ and $\boldsymbol{a}_{ik}, \boldsymbol{b}_{kj}$ is given by,

$$\frac{\delta \boldsymbol{o}_{ij}}{\delta \boldsymbol{a}_{ik}} = \frac{\delta \boldsymbol{o}_{ij}}{\delta \boldsymbol{b}_{kj}} = \frac{\exp\left(\boldsymbol{a}_{ik} + \boldsymbol{b}_{kj} - \boldsymbol{x}^{\star}\right)}{\sum_{k'} \exp\left(\boldsymbol{a}_{ik'} + \boldsymbol{b}_{k',j} - \boldsymbol{x}^{\star}\right)}$$
$$= \frac{\exp\left(\boldsymbol{a}_{ik} + \boldsymbol{b}_{kj} - \boldsymbol{x}^{\star}\right)}{\exp\left(\boldsymbol{o}_{ij} - \boldsymbol{x}^{\star}\right)} = \exp\left(\boldsymbol{a}_{ik} + \boldsymbol{b}_{kj} - \boldsymbol{o}_{ij}\right)$$

In the backward pass, we could recompute $\exp(\boldsymbol{a}_{ik} + \boldsymbol{b}_{kj} - \boldsymbol{o}_{ij})$ without the need to store $\exp(\boldsymbol{a}_{ik} + \boldsymbol{b}_{kj} - \boldsymbol{x}^{\star})$ in the forward pass, thus saving memory [4]. We found that this manual backpropagation led to a slight decrease in running speed but greatly increased memory savings, and thus use it for all our experiments.

**Speed comparison** Table 1 shows running speed and memory footprint measured under a single NVIDIA-A40 GPU, where we compare against the standard $\log$-sum-exp implementation of the inside algorithm which only leverages span-level

---

[4]This is also known as gradient checkpointing (Chen et al., 2016).

| Model | NT | ppl ($\downarrow$) |
|---|---|---|
| NHMM | 4096 | 147 |
| LHMM | 16384 | 131.8 |
| Rank HMM | 16384 | 127.0 |
| Rank HMM | 32768 | 126.4 |
| Rank PCFG† | 4096 | $174.5_{\pm 11.1}$ |
| Rank PCFG† | 8192 | $161.2_{\pm 8.9}$ |
| SN-PCFG | 4096 | $125.4_{\pm 4.1}$ |
| SN-PCFG | 8192 | $\mathbf{119.0}_{\pm 5.3}$ |

Table 2: Results on the PTB language modeling split from Mikolov et al. (2011). **NT** denotes the number of nonterminals and **ppl** denotes perplexity. Top results are from previous papers (Chiu et al., 2021; Yang et al., 2022), while the bottom results are from the current work. Our runs are averaged over 4 seeds.

parallelism (e.g., in Torch-Struct (Rush, 2020)). We can see that the use of log-einsum-exp trick significantly accelerate the running speed and reduce the memory footprint. FlashInside uses the kernel fusion and recomputation techniques in addition, resulting in further improvement, especially on larger grammars and longer sentences.

## 4 Experimental Setup

**Datasets.** We conduct experiments on the Penn Treebank (PTB) (Marcus et al., 1993) dataset with two different splits: one for language modeling (Mikolov et al., 2011), and one for unsupervised parsing (Shen et al., 2018, 2019). We also evaluate our model on Chinese Treebank 5.1 (CTB) (Xue et al., 2005) and German and French treebanks from SPRML (Seddah et al., 2014).

**Baselines.** Our HMM baselines include neural HMM (NHMM) (Chiu et al., 2021) , LHMM (Chiu et al., 2021), and Rank HMM (Yang et al., 2022). Our PCFG baselines include Neural/Compound PCFG (N/C-PCFG) (Kim et al., 2019), TN-PCFG (Yang et al., 2021b) and Rank PCFG (Yang et al., 2022). † denotes our reimplementation. For Rank PCFG we use rank size 4096. See Appendix B for more implementation details.

**Evaluation.** We use perplexity (ppl) to evaluate language modeling and sentence-level F1 (S-F1) (Kim et al., 2019) to evaluate unsupervised parsing.

## 5 Results

We compare our simple neural PCFG (SN-PCFG) to the baseline models. Table 2 shows the language modeling performance on PTB. SN-PCFG obtains significantly lower perplexity than Rank PCFG, and outperforms similarly-sized HMMs. This indicates that simple PCFGs provide a viable path

| Model | NT | Chinese | | French | | German | |
|---|---|---|---|---|---|---|---|
| | | S-F1(↑) | ppl(↓) | S-F1(↑) | ppl(↓) | S-F1(↑) | ppl(↓) |
| Left-Branching | - | 7.2 | - | 5.7 | - | 10.0 | |
| Right-Branching | - | 25.5 | - | 26.4 | - | 14.07 | - |
| Random Trees | - | 15.2 | - | 16.2 | - | 13.9 | - |
| Kim (2022) | - | - | - | 41.9 | - | 47.3 | - |
| Li and Lu (2023) | - | - | - | 48.7 | - | 40.8 | - |
| N-PCFG | 30 | $26.3_{\pm2.5}$ | | $45.0_{\pm2.0}$ | | $42.3_{\pm1.6}$ | |
| C-PCFG | 30 | $38.7_{\pm6.6}$ | - | $45.0_{\pm1.1}$ | - | $43.5_{\pm1.2}$ | - |
| TN-PCFG | 250 | $39.2_{\pm5.0}$ | | $39.1_{\pm4.1}$ | | $47.1_{\pm1.7}$ | |
| Rank PCFG | 4096 | $31.00_{\pm8.9}$ | $409.4_{\pm29.5}$ | $31.2_{\pm9.3}$ | $355.8_{\pm13.7}$ | $35.6_{\pm9.1}$ | $215.3_{\pm57.1}$ |
| Rank PCFG | 8192 | $32.4_{\pm8.2}$ | $372.6_{\pm31.4}$ | $32.9_{\pm10.6}$ | $332.2_{\pm60.8}$ | $38.9_{\pm9.6}$ | $190.5_{\pm65.9}$ |
| SN-PCFG | 4096 | $39.9_{\pm6.3}$ | $328.3_{\pm62.1}$ | $38.0_{\pm3.1}$ | $379.7_{\pm5.2}$ | $46.7_{\pm4.9}$ | $\mathbf{157.8}_{\pm65.6}$ |
| SN-PCFG | 8192 | $41.2_{\pm3.5}$ | $\mathbf{288.2}_{\pm11.7}$ | $43.3_{\pm9.9}$ | $\mathbf{259.9}_{\pm70.2}$ | $46.9_{\pm5.1}$ | $159.5_{\pm77.2}$ |
| SC-PCFG | 512 | $38.4_{\pm7.4}$ | - | $47.9_{\pm1.2}$ | - | $47.7_{\pm1.0}$ | - |
| SC-PCFG | 2048 | $\mathbf{42.9}_{\pm2.9}$ | - | $\mathbf{49.9}_{\pm1.7}$ | - | $\mathbf{49.1}_{\pm1.0}$ | - |

Table 3: Results on the Chinese, French, and German treebanks. All runs are averaged over 4 seeds.

| Model | NT | S-F1 (↑) | ppl (↓) |
|---|---|---|---|
| N-PCFG | 30 | 50.8 | 252.6 |
| C-PCFG | 30 | 55.2 | - |
| TN-PCFG | 500 | 57.7 | 210.0 |
| Rank PCFG | 4500 | 64.1 | 168.0 |
| Rank PCFG† | 4096 | $60.1_{\pm7.6}$ | $165.1_{\pm7.7}$ |
| Rank PCFG† | 8192 | $61.1_{\pm5.9}$ | $171.2_{\pm11.7}$ |
| N-PCFG† | 128 | $56.7_{\pm3.7}$ | $181.1_{\pm15.3}$ |
| SN-PCFG | 128 | $51.1_{\pm4.1}$ | $231.7_{\pm8.1}$ |
| SN-PCFG | 4096 | $\mathbf{65.1}_{\pm2.1}$ | $\mathbf{132.5}_{\pm4.9}$ |
| SN-PCFG | 8192 | $62.9_{\pm2.8}$ | $134.6_{\pm9.1}$ |
| SC-PCFG | 512 | $54.3_{\pm4.8}$ | - |
| SC-PCFG | 2048 | $60.6_{\pm3.6}$ | - |
| PRPN | - | 37.4 | - |
| ON | - | 47.7 | - |
| DIORA+span constraint | - | 61.2 | - |
| S-DIORA | - | 57.6 | - |
| Constituency test | - | 62.8 | - |
| StructFormer | - | 54.0 | - |
| Fast-R2D2 | - | 57.2 | - |
| Right-Branching | - | 39.5 | - |
| Oracle Trees | - | 84.3 | - |

Table 4: Unsupervised parsing performance on the PTB test set, including comparison against prior work (bottom): PRPN (Shen et al., 2018), ON (Shen et al., 2019), DIORA (Drozdov et al., 2019; Xu et al., 2021), S-DIORA (Drozdov et al., 2020), Constituency tests (Cao et al., 2020), StructFormer (Shen et al., 2021), and Fast-R2D2 (Hu et al., 2022). Whereever possible, we take the average F1 numbers across different seeds reported by the above papers (instead of the max).

towards scaling PCFGs, despite the strict independence assumption.

Table 4 and 3 show the unsupervised parsing performance. SN-PCFG consistently outperforms Rank PCFG in S-F1 while obtaining much lower perplexity. We also experiment with the compound version of simple PCFGs (SC-PCFG) which uses an auxiliary sentence-level vector to model sentence-level properties and uses variational inference for learning (see the appendix for the full parameterization). We find that SN-PCFG performs better on English while SC-PCFG achieves the best parsing performance in languages other than English. We remark that the compound parameterization is reported to be not compatible with low-rank parameterization probably due to optimization issues (Yang et al., 2021b). This work successfully scales compound PCFGs to thousands of states, which could be useful in some settings such as multimodal grammar induction which condition on vector representations of side information (Zhao and Titov, 2020; Jin and Schuler, 2020; Zhang et al., 2021, 2022; Li et al., 2022).

**Simple PCFG vs. Neural PCFG.** Despite the better scalablity of simple PCFGs, we find that under the same number of nonterminal (i.e., 128), SN-PCFG expectedly underperforms N-PCFG in both language modeling and unsupervised parsing (Table 4) due to the stronger independence assumption that is necessary for scaling. Nevertheless, N-PCFG does not scale well and (for example) runs into memory issues even with just 256 nonterminals, while SN-PCFG can scale to 8192 nonterminals on a single A40 GPU.

**Simple PCFG vs. Rank PCFG.** Recall that the rank PCFG and simple PCFG share an identical dynamic programming structure. The rank variable in the rank PCFG amounts to the nonterminal variable in the simple PCFG. Consequently, if we align the rank size in the rank PCFG with the nonterminal size in the simple PCFG, we achieve parity in terms of memory footprint and computational speed within the dynamic programming computation. In our experiments, we opt for a rank size of 4096 in the low-rank PCFG. The results, as presented in tables 2-4, showcase the worse perfor-

mance of the rank PCFG when compared to SN-PCFG with 4096 nonterminals. Interestingly, this work was motivated by our observation that merely augmenting the rank size of PCFG falls short in bridging the performance gap between HMMs and PCFGs in language modeling. This resulted in our exploring alternative parameterizations, culminating in the straightforward independent left/right productions based parameterization which yields superior results in both language modeling and unsupervised parsing.

## 6   Related Work

Independence assumptions are frequently made in grammar learning for tractability and scalability. Simple PCFGs assume independent generation of left and right children, thus resembling split-head dependency grammars (Eisner, 1996; Collins, 1997; Eisner and Satta, 1999; Paskin, 2001; Klein and Manning, 2004). We have shown that trading expressiveness (of grammar formalism) for scalablity is beneficial, and this idea could be applied to other complex grammar formalism of high parsing complexity, such as mildly context-sensitive grammars (Yang et al., 2023), synchronized grammars (Kim, 2021; Wang et al., 2022; Friedman et al., 2022; Lou and Tu, 2023) and lexicalized grammars (Zhu et al., 2020; Yang et al., 2021a).

## 7   Conclusion

In this work we explore a simpler variant of PCFGs (SimplePCFG) that shows better scaling properties than previous approaches in terms of both language modeling and unsupervised parsing performance. We also introduce a hardware-aware version of the inside algorithm (FlashInside) which improves over existing vectorized GPU implementations.

## Limitations

We have successfully bridged the gap between HMMs and PCFGs in language modeling. However, a significant disparity remains between PCFGs and neural models like Transformers. While we recognize the potential of our hardware-efficient inside algorithm implementation for conducting large-scale language modeling experiments, our aim is not to position PCFGs as direct rivals to neural models, given the intrinsic limitations arising from PCFG's strong context-free independence assumption. Our main objective is to enhance unsupervised PCFG learning, with a cen-

tral focus on optimizing the sentence log marginal likelihood objective function.

Simple PCFGs, due to their restrictive grammar nature, require many nonterminals for optimal performance. However, we observe diminishing returns while scaling up simple PCFGs. This phenomena is common in scaling up latent-variable models and future work might consider leveraging the technique from Liu et al. (2023) to mitigate this issue.

When scaling up simple PCFGs, the computation of grammar rule probabilities could also be expensive, especially when constructing the emission probability matrix of size $\mathbb{R}^{|\mathcal{P}| \times |\mathcal{V}|}$. Compound parameterization exacerbates this issue since each sentence will have its own set of grammar rule probabilities. Consequently, we only used up to 2048 nonterminals in our SC-PCFG experiments.

## Acknowlgedment

This study was supported by the National Natural Science Foundation of China (61976139) and by funds from an MIT-IBM Watson AI Lab grant.

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

## A   Neural Parameterization

We present the neural parameterization of our simple neural pcfg and simple compound pcfg. We use $\boldsymbol{E}_{\mathcal{G}} = \{\boldsymbol{w}_N | N \in \{\mathcal{S}\} \cup \mathcal{N} \cup \mathcal{P}\}$ to denote symbol embeddings for simple PCFG and use function $g_r(\cdot; \theta) = \pi_r$ parameterized by $\theta$ to denote neural parameterization function.

**Simple Neural PCFG**   We use neural networks to parameterize these rule probabilities. The neural parameterization of all rule probabilities $\pi_r$ starts from corresponding symbol embeddings in $\boldsymbol{E}_{\mathcal{G}}$. Parameterization function $g_r(\cdot; \theta)$ can be formulated as $g_r(\boldsymbol{E}_{\mathcal{G}}; \theta)$ in simple neural PCFG, which takes from one of these forms:

$$\pi_{S \to A} = \frac{\exp\left(\mathbf{u}_A^\top f_1\left(\mathbf{w}_S\right)\right)}{\sum_{A' \in \mathcal{N}} \exp\left(\mathbf{u}_{A'}^\top f_1\left(\mathbf{w}_S\right)\right)},$$

$$\pi_{B \frown A} = \frac{\exp\left(f_2\left(\mathbf{w}_B^\top\right) f_3\left(\mathbf{w}_A\right)\right)}{\sum_{B' \in \mathcal{N} \cup \mathcal{P}} \exp\left(f_2\left(\mathbf{w}_{B'}^\top\right) f_3\left(\mathbf{w}_A\right)\right)},$$

$$\pi_{A \frown C} = \frac{\exp\left(f_4\left(\mathbf{w}_C^\top\right) f_3\left(\mathbf{w}_A\right)\right)}{\sum_{C' \in \mathcal{N} \cup \mathcal{P}} \exp\left(f_4\left(\mathbf{w}_{C'}^\top\right) f_3\left(\mathbf{w}_A\right)\right)},$$

$$\pi_{T \to w} = \frac{\exp\left(\mathbf{u}_w^\top f_5\left(\mathbf{w}_T\right)\right)}{\sum_{w' \in \Sigma} \exp\left(\mathbf{u}_{w'}^\top f_5\left(\mathbf{w}_T\right)\right)}$$

where $f_1, f_5$ are two-layer residual networks; $f_2, f_3, f_4$ are one-linear-layer with ReLU activation function and residual connections. We highlight the usefulness of sharing symbol embedding across different grammar rules; and the use of residual connections in $f_2, f_3, f_4$.

**Simple Compound PCFG**   Similar to compound PCFGs, we parameterize our simple compound PCFGs with a latent variable $\boldsymbol{z} \sim p(\boldsymbol{z})$. We replace three rule probabilities $\pi_{B \frown A}, \pi_{B \frown A}$, and $\pi_{T \to w}$ with $\pi_r = g_r(\boldsymbol{z}, \boldsymbol{E}_{\mathcal{G}}; \theta)$, while leaving the remaining rule probabilities as $g_r(\boldsymbol{E}_{\mathcal{G}}; \theta)$. The function $\pi_r = g_r(\boldsymbol{z}, \boldsymbol{E}_{\mathcal{G}}; \theta)$ can take one of the following forms:

$$\pi_{B \frown A} = \frac{\exp\left(f_2\left(\mathbf{w}_B^\top\right) f_3'\left([\mathbf{w}_A; \boldsymbol{z}]\right)\right)}{\sum_{B' \in \mathcal{N} \cup \mathcal{P}} \exp\left(f_2\left(\mathbf{w}_{B'}^\top\right) f_3'\left([\mathbf{w}_A; \boldsymbol{z}]\right)\right)},$$

$$\pi_{A \frown C} = \frac{\exp\left(f_4\left(\mathbf{w}_C^\top\right) f_3'\left([\mathbf{w}_A; \boldsymbol{z}]\right)\right)}{\sum_{C' \in \mathcal{N} \cup \mathcal{P}} \exp\left(f_4\left(\mathbf{w}_{C'}^\top\right) f_3'\left([\mathbf{w}_A; \boldsymbol{z}]\right)\right)},$$

$$\pi_{T \to w} = \frac{\exp\left(\mathbf{u}_w^\top f_5'\left([\mathbf{w}_T; \boldsymbol{z}]\right)\right)}{\sum_{w' \in \Sigma} \exp\left(\mathbf{u}_{w'}^\top f_5'\left([\mathbf{w}_T; \boldsymbol{z}]\right)\right)}$$

where $f_3', f_5'$ are neural networks which are similar to $f_3, f_5$ but with different input shape.

## B   Implementation Details

We follow Mikolov et al. (2011) to preprocess PTB language modeling split data. For other datasets, we use the preprocessing from Yang et al. (2021b).

We implement our model based on the codebase of Yang et al. (2022). And most hyperparameters follow their settings. We use Xavier normal initialization to initialize neural networks. Our model is optimized by Adam optimizer with $\beta_1 = 0.75, \beta_2 = 0.999$, and learning rate 0.002. All dimensions of symbol embeddings are set to 512. Our FlashInside is implemented with Triton (Tillet et al., 2019) [5], an open-source python-like GPU programming language. For the latent variable $\boldsymbol{z}$ in SC-PCFG, we follow the implementation of Kim et al. (2019) which applies a max-pooling layer over the hidden states of the BiLSTM to obtain sentence representation and generates 64-dimensional mean vectors $\boldsymbol{\mu}(\boldsymbol{w})$ and log-variances $\log \boldsymbol{\sigma}(\boldsymbol{w})$ by leveraging an affine layer.

We run all experiments on NVIDIA V100 and NVIDIA A40. All experimental results are averaged from four runs.

---

[5]https://github.com/openai/triton