# OpenReview forum: "Simple Hardware-Efficient PCFGs with Independent Left and Right Productions"
_EMNLP/2023/Conference — EMNLP 2023 Findings_

### Official Review · Reviewer_ZpL2 · 2023-08-03

**Soundness:** 3

**Excitement:**

2: Mediocre: This paper makes marginal contributions (vs non-contemporaneous work), so I would rather not see it in the conference.

**Paper Topic And Main Contributions:**

This short paper proposes a simple factoring of the binary production rules of a PCFG. Instead of generating both children together, each child is produced independently. I am somewhat surprised that this has not been proposed before.
The authors then additionally discuss efficient an implementation that is optimized for GPUs.
Finally, the paper presents perplexity results on the Penn Treebank. The experiments are very small scale. The proposed method outperforms regular PCFGs and similar-sized HMMs, but there are no comparisons to the state of the art (which achieves far stronger results).

**Reasons To Accept:**

- Simple and clean formulation.

**Reasons To Reject:**

- Fairly light on substance.
- Results are far from state-of-the-art and very small scale.

**Reproducibility:**

4: Could mostly reproduce the results, but there may be some variation because of sample variance or minor variations in their interpretation of the protocol or method.

**Reviewer Confidence:**

4: Quite sure. I tried to check the important points carefully. It's unlikely, though conceivable, that I missed something that should affect my ratings.

---

> ### Author Rebuttal · Authors · 2023-08-28
>
> Thanks for your review. We want to emphasize that our proposed simple PCFG achieves SOTA performance in terms of **unsupervised parsing** on multiple languages: English, German and French. We have incorporated many strong baselines for English and will include other strong baselines such as [1] for German and French.  To the best of our knowledge, rank PCFG, TN-PCFG, [1] achieved SOTA performance in English, German and French, respectively. Our simple PCFG outperforms these models and thereby achieving new SOTA.  For language modeling, as we will explain later, we deliberately do not compare our model to strong neural models such as Transformers.
>
> [1] https://aclanthology.org/2023.acl-long.285/
>
> With regard to the experimental scale, we observe that our setup is well-aligned with common practices in the literature of unsupervised parsing. While we recognize the potential of our hardware-efficient inside algorithm implementation for conducting large-scale language modeling experiments, it's important to clarify our objective. Our aim is not to position PCFGs as direct rivals to neural models, given the intrinsic limitations arising from PCFG's strong context-free independence assumption.
>
> Our primary goal is improving unsupervised PCFG learning. The crux of this lies in optimizing the objective function of sentence log marginal likelihood.  By pivoting from a low-rank parameterization to a more straightforward independent left/right productions based parameterization, we observe better scaling properties (i.e., better language modeling/unsupervised parsing performance as we increase the number of nonterminals).

---

### Official Review · Reviewer_KFL6 · 2023-08-05

**Soundness:** 4

**Excitement:**

4: Strong: This paper deepens the understanding of some phenomenon or lowers the barriers to an existing research direction.

**Paper Topic And Main Contributions:**

The paper proposes a simple PCFG formulation for unsupervised parsing with independent left and right products, which scales more effectively to large number of nonterminals. Their formulation resembles the low-rank PCFG but does not utilize the low-rank restriction, granting it greater expressive power. Additionally, the authors present a hardware-efficient implementation that leverages span-level parallelism, the log-einsum-exp trick, and kernel fusion to enhance speed and reduce memory requirements. The experiments conducted on English, Chinese, French, and German treebanks demonstrate that the proposed approach achieves significantly improved parsing and language modeling capabilities.

**Reasons To Accept:**

The paper is clearly written, offering a simple yet robust formulation and an efficient implementation supported by solid experiments for a short paper.

**Reasons To Reject:**

I believe this is a valuable paper that should be accepted. One comparison that is missing is how the simple PCFG formulation affects memory/speed compared to the rank PCFG formulation. Presumably, the more expressive formulation would require more memory, and it would be insightful to compare these two approaches at the same memory footprint.

**Reproducibility:**

4: Could mostly reproduce the results, but there may be some variation because of sample variance or minor variations in their interpretation of the protocol or method.

**Reviewer Confidence:**

3: Pretty sure, but there's a chance I missed something. Although I have a good feel for this area in general, I did not carefully check the paper's details, e.g., the math, experimental design, or novelty.

---

> ### Author Rebuttal · Authors · 2023-08-28
>
> Thank you for your insightful comment. In fact, we have already investigated “how the simple PCFG formulation affects memory/speed compared to the rank PCFG formulation” . It's worth noting that the rank PCFG and simple PCFG share an identical dynamic programming structure. The rank variable in the rank PCFG amounts to the nonterminal variable in the simple PCFG, as mentioned in Sect. 2. Consequently, if we align the rank size in the rank PCFG with the nonterminal size in the simple PCFG, we achieve parity in terms of memory footprint and computational speed within the dynamic programming computation. This particular aspect has been explored in our experiments. As indicated in lines 183-184, we opt for a rank size of 4096 in the rank PCFG. The results, as presented in tables 2-4, showcase the worse performance of the rank PCFG when compared to SN-PCFG with 4096 nonterminals.
>
> Lastly, we'd like to remark that this work was motivated by our observation that merely augmenting the rank size of PCFG falls short in bridging the performance gap between HMMs and PCFGs in language modeling. This resulted in our exploring alternative parameterizations, culminating in the straightforward independent left/right productions based parameterization which yields superior results in both language modeling and unsupervised parsing.

---

### Official Review · Reviewer_vYhR · 2023-08-05

**Soundness:** 3

**Excitement:**

2: Mediocre: This paper makes marginal contributions (vs non-contemporaneous work), so I would rather not see it in the conference.

**Paper Topic And Main Contributions:**

This paper proposes a GPU efficient variant of PCFGs
(SimplePCFG) that scales better than previous approaches, demonstrating improvements over previous proposals in both language modelling (zh, de, en) and unsupervised parsing (ptb). The main contributions are FlashInside -- an inside estimation which is GPU parallelizable on the span level and some other tricks to make it GPU efficient.

**Reasons To Accept:**

* Detailed descriptions of the tricks needed
* Well rounded experiments

**Reasons To Reject:**

* The main contribution is a very simple almost trivial trick which caches children and hence allows for span-level parallelism  which has been proposed before (Cohen et al., 2013; Yang et al., 2022); this paper implements it on GPUs. **(In response to confusion raised in the author response:)  This technique (FlashInside) along with SimplePCFG, that is, independent left and right productions give rise to the more GPU-efficient implementation that has been helpful on PCFG-based unsupervised parsing but not on language modelling in terms of PPL**
* Experiments are small scale
* Lacks comparisons with more recent and better baselines

**Reproducibility:**

4: Could mostly reproduce the results, but there may be some variation because of sample variance or minor variations in their interpretation of the protocol or method.

**Reviewer Confidence:**

3: Pretty sure, but there's a chance I missed something. Although I have a good feel for this area in general, I did not carefully check the paper's details, e.g., the math, experimental design, or novelty.

---

> ### Author Rebuttal · Authors · 2023-08-28
>
> Thank you for your comment. We would like to clarify a major misunderstanding of our main contribution.
> - First of all, our primary contribution lies in the identification of a more effective parameterization for PCFGs, one that offers superior scalability properties in both language modeling and unsupervised parsing. This primary contribution (described in section 2) is independent of FlashInside, our second contribution (section 3).
> - Second, within FlashInside, the cache trick is introduced as a background technique in section 3.1 and is certainly not our main contribution.
>
> With regard to the experimental scale, we observe that our setup is well-aligned with common practices in the literature of unsupervised parsing. While we recognize the potential of our hardware-efficient inside algorithm implementation for conducting large-scale language modeling experiments,our aim is **not** to position PCFGs as direct rivals to neural models, given the intrinsic limitations arising from PCFG's strong context-free independence assumption.
>
> Our primary goal is improving unsupervised PCFG learning. The crux of this lies in optimizing the objective function of sentence log marginal likelihood.  By pivoting from a low-rank parameterization to a more straightforward independent left/right productions based parameterization, we observe better scaling properties (i.e., better language modeling/unsupervised parsing performance as we increase the number of nonterminals).
>
>  Regarding the lack of more recent and better baselines, we believe we have compared our models to other strong unsupervised parsers for English. We will add more baselines such as [1, 2] for German and French. To the best of our knowledge, rank PCFG, TN-PCFG, [1] achieved SOTA performance before in English, German and French, respectively. Our simple PCFG outperforms these models and thereby achieving new SOTA in these languages.
> For language modeling, as we explained above, we deliberately do not compare our model to strong neural models.
>
> [1] https://aclanthology.org/2023.acl-long.285/
>
> [2] https://aclanthology.org/2022.coling-1.479/

---

### Meta-Review · Area_Chair_LqYY · 2023-09-19

**Recommendation:** 3

**Metareview:**

This short paper proposes a simple PCFG formalism, which produces each child independently in a PCFG production rule. This simplification scales more effectively to large number of nonterminals. Experimental results on four treebanks show significant improvements over low-rank PCFG with same number of nonterminals, and also better performance than other unsupervised parsing algorithms (e.g. C-PCFG, S-DIORA and ON). This paper also introduces a hardware-efficient implementation to speedup the computation with lower memory usage.  Despite the simplicity of this approach, as a short paper, the experimental results on both unsupervised parsing and language modeling should be beneficial to the community of unsupervised parsing.

---

### Decision · Program_Chairs · 2023-10-07

**Decision:**

Accept-Findings

**Comment:**

This short paper proposes a simple PCFG formalism, which produces each child independently in a PCFG production rule. This simplification scales more effectively to large number of nonterminals. Experimental results on four treebanks show significant improvements over low-rank PCFG with same number of nonterminals, and also better performance than other unsupervised parsing algorithms (e.g. C-PCFG, S-DIORA and ON). This paper also introduces a hardware-efficient implementation to speedup the computation with lower memory usage.  Despite the simplicity of this approach, as a short paper, the experimental results on both unsupervised parsing and language modeling should be beneficial to the community of unsupervised parsing.